# Convergence of AA-Iterative Algorithm for Generalized *α*-Nonexpansive Mappings with an Application

**Ismat Beg** [1,*]🆔**, Mujahid Abbas** [2,3]🆔 **and Muhammad Waseem Asghar** [2]

[1] Department of Mathematics and Statistical Sciences, Lahore School of Economics, Lahore 53200, Pakistan
[2] Department of Mathematics, Government College University, Lahore 54000, Pakistan
[3] Department of Medical Research, China Medical University, Taichung 40402, Taiwan
[*] Correspondence: ibeg@lahoreschool.edu.pk

**Abstract:** The aim of this paper is to approximate the fixed points of generalized *α*-nonexpansive mappings using *AA*-iterative algorithm. We establish some weak and strong convergence results for generalized *α*-nonexpansive mappings in uniformly convex Banach spaces. A numerical example is also given to show that the *AA*-iterative algorithm converges faster than some others algorithms for generalized *α*-nonexpansive mappings. Lastly, using the *AA*-iterative algorithm, we approximate the weak solution of delay composite functional differential equation of the Volterra–Stieltjes type.

**Keywords:** *AA*-iteration; generalized *α*-nonexpansive mapping; fixed point; Banach space

**MSC:** 47H10; 47H09

## 1. Introduction

Throughout this paper, $\mathbb{Z}^+$ denotes the set of all positive integers, $B$ a Banach space, $G$ a closed convex subset of $B$, $\varphi : G \to G$ a mapping and $F_{ix}(\varphi)$ the set of all fixed points of $\varphi$.

A mapping $\varphi : G \to G$ is said to be

(1)    A contraction if, for all $a, b \in G$, there exists $\alpha \in (0, 1)$ such that

$$\|\varphi(a) - \varphi(b)\| \leq \alpha \|a - b\|.$$

(2)    A nonexpansive mapping if

$$\|\varphi(a) - \varphi(b)\| \leq \|a - b\|,$$

holds for all $a, b \in G$.

(3)    Quasi-non-expansive if, for all $a \in G$ and $a^* \in F_{ix}(\varphi)$, we have

$$\|\varphi(a) - a^*\| \leq \|a - a^*\|.$$

Browder [1] showed that, if $B$ is a uniformly convex Banach space and $G$ is a nonempty closed convex subset of $B$, then a nonexpansive mapping on $G$ has a fixed point.

In 2008, Suzuki [2] introduced a new type of mapping satisfying Condition (C). A self mapping $\varphi$ on $G$ satisfies Condition (C) if for $a, b \in G$ with

$$\frac{1}{2}\|a - \varphi(a)\| \leq \|a - b\|,$$

we have

$$\|\varphi(a) - \varphi(b)\| \leq \|a - b\|, \tag{1}$$

The mappings satisfying Condition (C) do not need to be continuous; hence, Condition (C) is weaker than the one depicting nonexpansive mappings. However, mappings satis-

fying Condition (C) were stronger than the one defining quasi-non-expansive mappings. Suzuki [2] studied the existence and convergence results for such mappings.

In 2011, Aoyama and Kohshaka [3] defined a new class of mappings known as $\alpha$-nonexpansive mappings on normed spaces and studied its fixed points.

A mapping $\varphi : G \to G$ is $\alpha$-nonexpansive if, for $a, b \in G$ and $\alpha < 1$, the following holds:

$$\|\varphi(a) - \varphi(b)\|^2 \leq \alpha\|b - \varphi(a)\|^2 + \alpha\|a - \varphi(b)\|^2 + (1 - 2\alpha)\|a - b\|^2.$$

Clearly, for $\alpha = 0$, we have a class of nonexpansive mappings. An example of a discontinuous $\alpha$-nonexpansive mapping (with $\alpha > 0$) was given in [3], which shows that the class of $\alpha$-nonexpansive mappings was larger than the nonexpansive mappings (see also [4]).

Pant and Shukla [5] introduced a class of mapping (called the generalized $\alpha$-nonexpansive mapping) as follows:

For all $a, b \in G$, there exists $\alpha \in (0, 1)$, such that

$$\frac{1}{2}\|a - \varphi(a)\| \leq \|a - b\|$$

implies that

$$\|\varphi(a) - \varphi(b)\| \leq \alpha\|b - \varphi(a)\| + \alpha\|a - \varphi(b)\| + (1 - 2\alpha)\|a - b\|,$$

Many researchers studied the approximation of fixed points of such mappings in Banach spaces. For instance, we refer to [5–8].

The following example in [4] shows that the generalized $\alpha$-nonexpansive mapping needs not satisfy Condition (C).

**Example 1.** *Let set $G = [0, \infty)$ be equipped with usual norm $|.|$. Define $\varphi : G \to G$ by:*

$$\varphi(a) = \begin{cases} \frac{a}{2}, & \text{if } a > 2 \\ 0, & \text{if } a \in [0, 2]. \end{cases}$$

*$\varphi$ satisfies Condition $(C_\alpha)$, but $\varphi$ is not a nonexpansive mapping.*

Banach [9] proved that fixed points of contraction mappings can be approximated with the Picard iterative algorithm [10]. The Picard sequence $\{a_n\}$ is defined as follows:

$$\begin{cases} a_1 \in G, \\ a_{n+1} = \varphi(a_n) \ n \in \mathbb{Z}^+. \end{cases} \tag{2}$$

The above sequence generated by the Picard algorithm does not converge to a fixed point of nonexpansive mappings. For more details, we refer to [11].

In 1953, Mann [12] introduced a new iterative algorithm to approximate a fixed point for nonexpansive mappings. The sequence obtained by this algorithm is defined as follows:

$$\begin{cases} a_1 \in G, \\ a_{n+1} = (1 - \eta_n)a_n + \eta_n\varphi(a_n) \ n \in \mathbb{Z}^+, \end{cases} \tag{3}$$

where $\{\eta_n\}$ is an appropriate sequence in $(0, 1)$.

The Mann iteration failed to approximate the fixed point in the case of pseudocontractive mapping. To overcome this problem, Ishikawa [13] introduced a two-step iterative algorithm to approximate the fixed point of pseudocontractive mapping.

Sequence $\{a_n\}$, obtained by Ishikawa algorithm, is given as follows:

$$\begin{cases} a_1 \in G \\ a_{n+1} = (1 - \eta_n)a_n + \eta_n \varphi(b_n) \\ b_n = (1 - \rho_n)a_n + \rho_n \varphi(a_n) \quad n \in \mathbb{Z}^+, \end{cases} \quad (4)$$

where $\{\eta_n\}$ and $\{\rho_n\}$ are sequences in $(0,1)$.

Noor [14] in 2000, Agarwal et al. [15] in 2007, Abbas and Nazir [16] in 2014, Thakur et al. [17] in 2017, and Ullah and Arshad [18] in 2018 proposed different iterative algorithms (see Table 1): let $a_1 \in G$ be an initial guess.

**Table 1.** Different Iterative Algorithms.

| Name | Algorithms |
|:---:|:---|
| Noor | $a_{n+1} = (1 - \eta_n)a_n + \eta_n \varphi(b_n)$<br>$b_n = (1 - \rho_n)a_n + \rho_n \varphi(c_n)$<br>$c_n = (1 - \sigma_n)a_n + \sigma_n \varphi(a_n)$ |
| Agarwal et al. | $a_{n+1} = (1 - \eta_n)\varphi(a_n) + \eta_n \varphi(b_n)$<br>$b_n = (1 - \rho_n)a_n + \rho_n \varphi(a_n)$ |
| Abbas et al. | $a_{n+1} = (1 - \eta_n)\varphi(b_n) + \eta_n \varphi(c_n)$<br>$b_n = (1 - \rho_n)\varphi(a_n) + \rho_n \varphi(c_n)$<br>$c_n = (1 - \sigma_n)a_n + \sigma_n \varphi(a_n)$ |
| Thakur et al. | $a_{n+1} = (1 - \eta_n)\varphi(c_n) + \eta_n \varphi(b_n)$<br>$b_n = (1 - \rho_n)c_n + \rho_n \varphi(c_n)$<br>$c_n = (1 - \sigma_n)a_n + \sigma_n \varphi(a_n)$ |
| Ullah et al. | $a_{n+1} = \varphi(b_n)$<br>$b_n = \varphi(c_n)$<br>$c_n = (1 - \eta_n)a_n + \eta_n \varphi(a_n)$ |

Where $\{\eta_n\}$, $\{\rho_n\}$ and $\{\sigma_n\}$ are the sequences of parameters in $(0,1)$.

Recently, Abbas et al. [19] introduced a new iterative algorithm known as the *AA*-iterative algorithm, which converges faster than the iterative algorithms mentioned above for the class of enriched contraction and contraction mapping. The sequence defined by this algorithm is given as follows:

$$\begin{cases} a_1 \in G \\ a_{n+1} = \varphi(b_n) \\ b_n = \varphi((1 - \eta_n)\varphi(d_n) + \eta_n \varphi(c_n)) \\ c_n = \varphi((1 - \rho_n)d_n + \rho_n \varphi(d_n)) \ n \in \mathbb{Z}^+ \\ d_n = (1 - \sigma_n)a_n + \sigma_n \varphi(a_n), \end{cases} \quad (5)$$

where $\{\eta_n\}$, $\{\rho_n\}$ and $\{\sigma_n\}$ are sequence in $(0,1)$.

Using the Ishikawa algorithm, Phuengratta [20] in 2011 proved te convergence results for Suzuki-type generalized nonexpansive mappings. In 2019, Ali et al. [21] employed an iterative algorithm in [17] to prove the convergence results for Suzuki-type generalized nonexpansive mapping in uniformly convex Banach spaces. The fixed-point theorems for Suzuki-type generalized nonexpansive mapping and some other nonlinear mappings were studied by many researchers [22–24]. Hence, the approximation of the fixed point of a more general class of mappings in fewer steps has been a matter of great interest for many authors due to its theoretical and practical applications. This is the main motivation of this paper.

Motivated by the work in [20,21], we prove some strong and weak convergence results by using the *AA*-iterative algorithm (5) for the generalized $\alpha$-nonexpansive mappings in

uniformly convex Banach spaces. Our work is more general and unifies the comparable results in the existing literature, for instance, the results given in [17,21].

## 2. Preliminaries

**Definition 1** ([21]). *Let G be a nonempty closed convex subset of a Banach space B. A mapping $\varphi : G \to G$ is called demiclosed with respect to $b \in B$ if, for each sequence $\{a_n\}$ in G and $a \in G$, $\{a_n\}$ converges weakly to a, and $\{\varphi(a_n)\}$ converges strongly to b, implying that $\varphi(a) = b$.*

**Definition 2** ([25]). *A Banach space B satisfies Opial's condition if, for each sequence $\{a_n\}$ converging weakly to $a \in B$, the following holds:*

$$\liminf_{n \to \infty} \|a_n - a\| < \liminf_{n \to \infty} \|a_n - b\|,$$

*for for all $b \in B$ with $a \neq b$.*

Sentor and Dotson [26] introduced the concept of mapping satisfying Condition (I), which is defined as follows:

**Definition 3.** *A mapping $\varphi : G \to G$ satisfies Condition (I) if there exists an increasing function $r : [0, \infty) \to [0, \infty)$ with $r(0) = 0$ and $r(t) > 0$, for all $t > 0$, such that*

$$d(a, \varphi(a)) \geq r(d(a, F_{ix}(\varphi))), \text{ for all } a \in G,$$

*where $d(a, F_{ix}(\varphi)) = \inf\{d(a, a^*) : a^* \in F_{ix}(\varphi)\}$.*

**Definition 4.** *Let $\{a_n\}$ be a bounded sequence in a Banach space B. Define a mapping $r(\cdot, \{a_n\}) : B \to \mathbb{R}^+$ by*

$$r(a, \{a_n\}) = \limsup_{n \to \infty} \|a_n - a\|.$$

*For each $a \in B$, value $r(a, \{a_n\})$ is called the asymptotic radius of $\{a_n\}$ at a.*

*The asymptotic radius of $\{a_n\}$ relative to $G \subset B$ is defined as follows:*

$$r(G, \{a_n\}) = \inf\{r(a, \{a_n\}) : a \in G\}.$$

*The asymptotic center of $\{a_n\}$ relative to G is the set*

$$A(G, \{a_n\}) = \{a \in G : r(G, \{a_n\}) = r(a, \{a_n\})\}.$$

The asymptotic center of $\{a_n\}$ with respect to G is nonempty and convex whenever G is weakly compact [27,28]. Moreover, set $A(G, \{a_n\})$ is a singleton, provided that B is a uniformly convex Banach space [29].

**Proposition 1** ([5]). *Every mapping satisfying Condition (C) is generalized $\alpha$-nonexpansive mapping, but the converse does not hold in general.*

**Proposition 2** ([5]). *Let G be a nonempty subset of a Banach space B and $\varphi : G \to G$ a generalized $\alpha$-nonexpansive mapping . Then, for all $a, b \in G$, we have*

$$\|a - \varphi(a)\| \leq \frac{(3 + \alpha)}{(1 - \alpha)} \|a - \varphi(a)\| + \|a - b\|.$$

**Theorem 1** ([2]). *Let G be a weakly compact convex subset of a uniformly Banach space B and a mapping $\varphi$ on G satisfies Condition (C). Then, $\varphi$ has a fixed point.*

**Lemma 1** ([30]). *Let B be a uniformly convex Banach space and $0 < \rho_n < 1$ for all $n \in \mathbb{Z}^+$. Let $\{a_n\}$ and $\{b_n\}$ be the two sequences such that $\limsup_{n\to\infty} \|a_n\| \leq d$, $\limsup_{n\to\infty} \|b_n\| \leq d$ and $\limsup_{n\to\infty} \|(1-\rho_n)a_n + \rho_n b_n\| = d$ holds for some $d \geq 0$ then $\lim_{n\to\infty} \|a_n - b_n\| = 0$.*

**Lemma 2** ([5]). *Let $\varphi : G \to G$ be a generalized $\alpha-$ nonexpansive mapping that satisfies Opial's property. If $\{a_n\}$ converges weakly to c and $\lim_{n\to\infty} \|a_n - \varphi(a_n)\| = 0$, then $\varphi(c) = c$, that is $I - \varphi$ is demiclosed at zero, where I is an identity mapping on B.*

**Proposition 3** ([31]). *Let $\varphi : G \to G$ be a generalized $\alpha$-nonexpansive mapping; then, the following holds.*

*(i)*    *If $\varphi$ satisfies Condition (C), then $\varphi$ satisfies Condition ($C_\alpha$).*
*(ii)*   *If $\varphi$ satisfies Condition ($C_\alpha$) and $F_{ix}(\varphi) \neq \emptyset$, then $\varphi$ is quasi-non-expansive.*

## 3. Convergence Analysis

In this section, we prove some strong and weak convergence results using $AA$-iterative scheme (5) for generalized $\alpha$-nonexpansive mappings in a uniformly convex Banach space $B$, and all the results in this section generalize the corresponding results of Thakur et al. [17] and Ali et al. [21].

**Lemma 3.** *Let G be a nonempty closed convex subset of a uniformly convex Banach space B and $\varphi : G \to G$ a generalized $\alpha$-nonexpansive mapping with $F_{ix}(\varphi) \neq \emptyset$. If $\{a_n\}$ is a sequence defined by $AA$-iterative algorithm (5), then $\lim_{n\to\infty} \|a_n - a^*\|$ exists for all $a^* \in F_{ix}(\varphi)$.*

**Proof.** Let $a^* \in F_{ix}(\varphi)$. Since $\varphi$ satisfies Condition ($C_\alpha$), with Proposition 3, $\varphi$ is quasi-nonexpansive mapping, that is,

$$\|\varphi(a) - a^*\| \leq \|a - a^*\|.$$

Using Iterative Algorithm (5), we have

$$\begin{aligned}
\|d_n - a^*\| &= \|(1-\sigma_n)a_n + \sigma_n \varphi(a_n) - a^*\| \\
&\leq (1-\sigma_n)\|a_n - a^*\| + \sigma_n\|\varphi(a_n) - a^*\|.
\end{aligned} \tag{6}$$

As $\varphi$ is generalized $\alpha$ nonexpansive mapping with $\varphi(a^*) = a^*$, we have

$$\begin{aligned}
\|\varphi(a_n) - a^*\| &\leq \alpha\|a^* - \varphi(a_n)\| + \alpha\|a_n - \varphi(a^*)\| + (1-2\alpha)\|a_n - a^*\| \\
&\leq \alpha\{\|a^* - \varphi(a^*)\| + \|\varphi(a_n) - \varphi(a^*)\|\} + \alpha\|a_n - \varphi(a^*)\| \\
&\quad + (1-2\alpha)\|a_n - a^*\| \\
&\leq \|a_n - a^*\|.
\end{aligned} \tag{7}$$

Using (7) in (6), we obtain that

$$\begin{aligned}
\|d_n - a^*\| &\leq (1-\sigma_n)\|a_n - a^*\| + \sigma_n\|a_n - a^*\| \\
&= \|a_n - a^*\|.
\end{aligned} \tag{8}$$

If $t_n = (1-\rho_n)d_n + \rho_n \varphi(d_n)$, then

$$\|c_n - a^*\| = \|\varphi(t_n) - a^*\|. \tag{9}$$

Now,

$$\begin{aligned}
\|\varphi(t_n) - \varphi(a^*)\| &\leq \alpha\|a^* - \varphi(t_n)\| + \alpha\|t_n - \varphi(a^*)\| + (1-2\alpha)\|t_n - a^*\| \\
&\leq \|t_n - a^*\|.
\end{aligned} \tag{10}$$

In addition,

$$\begin{aligned}
\|t_n - a^*\| &\leq \|(1 - \rho_n)d_n + \rho_n \varphi(d_n) - a^*\| \\
&\leq (1 - \rho_n)\|d_n - a^*\| - \rho_n \|\varphi(d_n) - a^*\|,
\end{aligned} \tag{11}$$

and

$$\begin{aligned}
\|\varphi(d_n) - a^*\| &\leq \alpha\|a^* - d_n\| + \alpha\|d_n - \varphi(a^*)\| + (1 - 2\alpha)\|d_n - a^*\| \\
&\leq \|d_n - a^*\|.
\end{aligned} \tag{12}$$

Putting (8) and (12) in (11), we obtain

$$\|t_n - a^*\| \leq \|a_n - a^*\|. \tag{13}$$

From (10) and (13), we have

$$\|\varphi(t_n) - a^*\| \leq \|a_n - a^*\|. \tag{14}$$

It follows from (9) and (14) that

$$\|c_n - a^*\| \leq \|a_n - a^*\|. \tag{15}$$

Now, take $u_n = (1 - \eta_n)\varphi d_n + \eta \varphi(c_n)$,

$$\begin{aligned}
\|b_n - a^*\| &\leq \|\varphi(u_n) - a^*\| \leq \|u_n - a^*\| \\
&\leq \alpha\|a^* - \varphi(u_n)\| + \alpha\|u_n - \varphi(a^*)\| + (1 - 2\alpha)\|u_n - a^*\| \\
&\leq \alpha\|\varphi(u_n) - (a^*)\| + (1 - \alpha)\|u_n - a^*\| \\
&\leq \|u_n - a^*\|.
\end{aligned} \tag{16}$$

$$\begin{aligned}
\|u_n - a^*\| &\leq \|(1 - \eta_n)\varphi(d_n) + \eta \varphi(c_n) - a^*\| \\
&\leq (1 - \eta_n)\|\varphi(d_n) - a^*\| + \eta_n \|\varphi(c_n) - a^*\| \\
&\leq (1 - \eta_n)\|\varphi(d_n) - a^*\| + \eta_n \|\varphi(c_n) - a^*\|,
\end{aligned} \tag{17}$$

and

$$\begin{aligned}
\|\varphi(c_n) - a^*\| &\leq \alpha\|a^* - \varphi(c_n) + \|(c_n) - \varphi(a^*)\| + (1 - 2\alpha)\|c_n - a^*\| \\
&\leq \alpha\|\varphi(c_n) - (a^*)\| + (1 - \alpha)\|c_n - a^*\| \\
&\leq \|c_n - a^*\|.
\end{aligned} \tag{18}$$

Using (12) and (18) in (17), we obtain

$$\|u_n - a^*\| \leq \|a_n - a^*\|. \tag{19}$$

Putting (19) in (16), we obtain

$$\|b_n - a^*\| \leq \|a_n - a^*\|. \tag{20}$$

Now,

$$\|a_{n+1} - a^*\| \leq \|\varphi(b_n) - a^*\|, \tag{21}$$

and

$$\begin{aligned}
\|\varphi(b_n) - a^*\| \leq &\alpha\|a^* - \varphi(b_n)\| + \alpha\|b_n - \varphi(a^*)\| + (1 - 2\alpha)\|b_n - a^*\| \\
\leq &\alpha\|\varphi(b_n) - \varphi(a^*)\| + (1 - \alpha)\|b_n - a^*\| \\
\leq &\|b_n - a^*\|.
\end{aligned} \tag{22}$$

Putting (22) in (21), we have

$$\|a_{n+1} - a^*\| \leq \|b_n - a^*\|. \tag{23}$$

From (20) and (23), we obtain that

$$\|a_{n+1} - a^*\| \leq \|a_n - a^*\|. \tag{24}$$

This shows that $\{\|a_n - a^*\|\}$ is decreasing and bounded from the below sequence for each $a^* \in F_{ix}(\varphi)$.

Hence, $\lim_{n\to\infty}\|a_n - a^*\|$ exists. $\quad\square$

**Lemma 4.** *Let G be a nonempty closed convex subset of a uniformly convex Banach space B and $\varphi : G \to G$ a generalized $\alpha$-nonexpansive mapping. If $\{a_n\}$ is a sequence defined by AA-iterative algorithm (5), then $F_{ix}(\varphi) \neq \varnothing$ if and only if $\{a_n\}$ is bounded and $\lim_{n\to\infty}\|a_n - \varphi(a_n)\| = 0$.*

**Proof.** With Lemma 3 above, $\lim_{n\to\infty}\|a_n - a^*\|$ exists, and $\{a_n\}$ is bounded. Put

$$\lim_{n\to\infty}\|a_n - a^*\| = k. \tag{25}$$

From (15), (18), (20) and (25), we have

$$\limsup_{n\to\infty}\|d_n - a^*\| \leq \limsup_{n\to\infty}\|a_n - a^*\| \leq k, \tag{26}$$

$$\limsup_{n\to\infty}\|c_n - a^*\| \leq \limsup_{n\to\infty}\|a_n - a^*\| \leq k, \tag{27}$$

$$\limsup_{n\to\infty}\|b_n - a^*\| \leq \limsup_{n\to\infty}\|a_n - a^*\| \leq k. \tag{28}$$

It follows from (7) that

$$\|\varphi(a_n) - a^*\| = \|\varphi(a_n) - \varphi(a^*)\| \leq \|a_n - a^*\|.$$
$$\limsup_{n\to\infty}\|\varphi(a_n) - a^*\| \leq k. \tag{29}$$

Thus,

$$\|a_{n+1} - a^*\| = \|\varphi(b_n) - \varphi(a^*)\| \leq \|b_n - a^*\|. \tag{30}$$

By taking lim inf as $n \to \infty$, we obtain

$$k \leq \liminf_{n\to\infty}\|b_n - a^*\|. \tag{31}$$

From (28) and (31), we have

$$\lim_{n\to\infty}\|b_n - a^*\| = k. \tag{32}$$

Now, from (30), we obtain that

$$\|a_{n+1} - a^*\| \leq \|b_n - a^*\| \leq \|\varphi(c_n) - a^*\| \leq \|c_n - a^*\|, \tag{33}$$

which, on taking lim inf as $n \to \infty$, gives that

$$k \leq \liminf_{n \to \infty} \|c_n - a^*\|. \tag{34}$$

With (27) and (34), we obtain

$$\lim_{n \to \infty} \|c_n - a^*\| = k.$$

From (33), we have

$$\|a_{n+1} - a^*\| \leq \|c_n - a^*\| \leq \|\varphi(c_n) - a^*\| \leq \|d_n - a^*\|. \tag{35}$$

On taking lim inf as $n \to \infty$, we obtain that

$$k \leq \liminf_{n \to \infty} \|d_n - a^*\|. \tag{36}$$

Thus, from (25) and (36), we obtain

$$\lim_{n \to \infty} \|d_n - a^*\| = k.$$

In addition,

$$
\begin{aligned}
k &\leq \lim_{n \to \infty} \|d_n - a^*\| \\
&= \lim_{n \to \infty} \|(1 - \sigma_n)a_n + \sigma_n \varphi(a_n)) - a^*\| \\
&\leq \lim_{n \to \infty} (1 - \sigma_n)\|a_n - a^*\| + \sigma_n \|\varphi(a_n) - a^*\| \\
&\leq \lim_{n \to \infty} (1 - \sigma_n)\|a_n - a^*\| + \sigma_n \|a_n - a^*\| \\
&\leq \lim_{n \to \infty} \|a_n - a^*\| \\
&\leq k.
\end{aligned}
$$

Hence,

$$\lim_{n \to \infty} \|(1 - \sigma_n)(a_n - a^*) + \sigma_n(\varphi(a_n) - a^*)\| = k. \tag{37}$$

From (25), (29), (37) and Lemma 1, we obtain

$$\lim_{n \to \infty} \|a_n - \varphi(a_n)\| = 0.$$

*Conversely*, suppose $\{a_n\}$ is bounded and $\lim_{n \to \infty} \|a_n - \varphi(a_n)\| = 0$.
Let $a^* \in A(G, \{a_n\})$. Through Proposition 2, we have

$$
\begin{aligned}
r(\varphi, \{a_n\}) &= \limsup_{n \to \infty} \|a_n - \varphi(a^*)\| \\
&\leq \limsup_{n \to \infty} (\frac{(3 + \alpha)}{(1 - \alpha)} \|a_n - \varphi(a_n) + \|a_n - a^*\|) \\
&\leq \limsup_{n \to \infty} \|a_n - a^*\| \\
&= r(a^*, \{a_n\}) = r(G, \{a_n\}),
\end{aligned}
$$

which implies that $\varphi(a^*) \in A(G, \{a_n\})$.
Since $B$ is uniformly convex, $A(G, \{a_n\})$ is a singleton.
Hence, we have $\varphi(a^*) = a^*$. $\quad \square$

**Theorem 2.** *Let $G$ be a nonempty closed convex subset of a uniformly convex Banach space $B$ and $\varphi : G \to G$ a generalized $\alpha$-nonexpansive mapping. If $\{a_n\}$ is a sequence defined by the*

*AA-iterative algorithm* (5), *then* $\{a_n\}$ *converges weakly to a point of* $F_{ix}(\varphi)$, *provided that B satisfies Opial's condition.*

**Proof.** Let $a^* \in F_{ix}(\varphi)$. Through Lemma 3, $\lim_{n\to\infty} \|a_n - a^*\|$ exists. Now, we show that $\{a_n\}$ has a unique weak subsequential limit in $F_{ix}(\varphi)$.

Suppose $a$ and $b$ are weak limits of subsequences $\{a_{n_i}\}$ and $\{a_{n_j}\}$ of $\{a_n\}$, respectively. From Lemma 4, we have $\lim_{n\to\infty} \|a_n - \varphi(a_n)\| = 0$. Moreover, from Lemma 2, $I - \varphi$ is demiclosed at zero.

This implies that $(I - \varphi)a = 0$, that is, $a = \varphi(a)$. Similarly, $b = \varphi(b)$.

Now, we show the uniqueness. If $a \neq b$,, then by using Opial's condition, we have

$$
\begin{aligned}
\lim_{n\to\infty} \|a_n - a\| &= \lim_{n_i\to\infty} \|a_{n_i} - a\| \\
&< \lim_{n_i\to\infty} \|a_{n_i} - b\| \\
&= \lim_{n\to\infty} \|a_n - b\| \\
&= \lim_{n_j\to\infty} \|a_{n_j} - b\| \\
&< \lim_{n_j\to\infty} \|a_{n_j} - a\| \\
&= \lim_{n\to\infty} \|a_n - a\|,
\end{aligned}
$$

a contradiction; so, $a = b$. Consequently, $\{a_n\}$ converges weakly to a point of $F_{ix}(\varphi)$. □

**Theorem 3.** *Let G be a nonempty closed convex subset of a uniformly convex Banach space B and* $\varphi : G \to G$ *a generalized $\alpha$-nonexpansive mapping. If* $\{a_n\}$ *is a sequence defined by AA-iterative algorithm* (5), *then* $\{a_n\}$ *converges to a point of* $F_{ix}(\varphi)$ *if and only if* $\liminf_{n\to\infty} d(a_n, F_{ix}(\varphi)) = 0$ *or* $\limsup_{n\to\infty} d(a_n, F_{ix}(\varphi)) = 0$, *where* $d(a_n, F_{ix}(\varphi)) = \inf\{\|a_n - a^*\| : a^* \in F_{ix}(\varphi)\}$.

**Proof.** If $\{a_n\}$ converges to a fixed point $a^* \in F_{ix}(\varphi)$, then obviously, we have $\liminf_{n\to\infty} d(a_n, F_{ix}(\varphi)) = 0$ and $\limsup_{n\to\infty} d(a_n, F_{ix}(\varphi)) = 0$.

*Conversely*, suppose that $\liminf_{n\to\infty} d(a_n, F_{ix}(\varphi)) = 0$. From Lemma 3, $\lim_{n\to\infty} \|a_n - a^*\|$ exists for all $a^* \in F_{ix}(\varphi)$. Thus, by assumption,

$$
\lim_{n\to\infty} d(a_n, F_{ix}(\varphi)) = 0.
$$

We now show that $\{a_n\}$ is a Cauchy sequence in $G$. As $\lim_{n\to\infty} d(a_n, F_{ix}(\varphi)) = 0$, for given $\varepsilon > 0$, there exists $m_0 \in Z^+$, such that, for all $n \geq m_0$,

$$
d(a_n, F_{ix}(\varphi)) < \frac{\varepsilon}{2},
$$

that is

$$
\inf\{\|a_n - a^*\| : a^* \in F_{ix}(\varphi)\} < \frac{\varepsilon}{2}.
$$

In particular, $\inf\{\|a_n - a^*\| : a^* \in F_{ix}(\varphi)\} < \frac{\varepsilon}{2}$. Therefore, there exists $a^* \in F_{ix}(\varphi)$ such that

$$
\|a_{m_0} - a^*\| < \frac{\varepsilon}{2}.
$$

Now, for $m, n \geq m_0$,

$$
\begin{aligned}
\|a_{n+m} - a_n\| &\leq \|a_{m+n} - a^*\| + \|a_n - a^*\| \\
&\leq \|a_{m_0} - a^*\| + \|a_{m_0} - a^*\| \\
&= 2\|a_{m_0} - a^*\| \\
&< \varepsilon.
\end{aligned}
$$

This shows that $\{a_n\}$ is a Cauchy sequence in $G$. As $G$ is a closed subset of a Banach space $B$, there is a point $r \in G$, such that $\lim_{n \to \infty} a_n = r$. Now, $\lim_{n \to \infty} d(a_n, F_{ix}(\varphi)) = 0$ gives that $\lim_{n \to \infty} d(a_n, F_{ix}(\varphi)) = 0$. Hence, $r \in F_{ix}(\varphi)$. $\square$

**Theorem 4.** *Let $G$ be a nonempty compact convex subset of a uniformly convex Banach space $B$ and $\varphi : G \to G$ be a generalized $\alpha$-nonexpansive mapping. If $\{a_n\}$ is a sequence defined with the $AA$-iterative algorithm* (5)*, then $\{a_n\}$ converges strongly to a fixed point of $\varphi$.*

**Proof.** From Theorem 1, $F_{ix}(\varphi) \neq \varnothing$; so, via Lemma 4, we have

$$
\lim_{n \to \infty} \|a_n - \varphi(a_n)\| = 0.
$$

Since $G$ is compact, there is a subsequence $\{a_{n_k}\}$ of $\{a_n\}$, such that $a_{n_k} \to a^*$ for some $a^* \in G$. Through Proposition 2, we have

$$
\|a_{n_k} - \varphi(a^*)\| \leq \frac{(3+\alpha)}{(1-\alpha)} \|a_{n_k} - \varphi(a_{n_k})\| + \|a_{n_k} - a^*\| \quad \forall \quad k \geq 1.
$$

On taking the limit to be $k \to \infty$, we obtain $a_{n_k} \to \varphi(a^*)$. This implies that $\varphi(a^*) = a^*$, that is, $a^* \in F_{ix}(\varphi)$.

In addition, $\lim_{n \to \infty} \|a_n - a^*\|$ exists by Lemma (3). Thus, $a^*$ is the limit of a sequence $\{a_n\}$. $\square$

Now, we prove a strong convergence result using Condition (I).

**Theorem 5.** *Let $G$ be a nonempty closed and convex subset of a uniformly convex Banach space $B$ and $\phi : G \to G$ be generalized $\alpha$-nonexpansive mapping satisfying Condition (I). Then, sequence $\{a_n\}$, defined with $AA$-iterative Algorithm* (5)*, converges strongly to a fixed point of $\varphi$.*

**Proof.** As proven in Lemma 4,

$$
\lim_{n \to \infty} \|a_n - \varphi(a_n)\| = 0. \tag{38}
$$

From Condition (I) and (31), we obtain

$$
0 \leq \lim_{n \to \infty} r(d(a_n, F_{ix}(\varphi))) \leq \lim_{n \to \infty} \|a_n - \varphi(a_n)\|,
$$

which implies

$$
\lim_{n \to \infty} r(d(a_n, F_{ix}(\varphi))) = 0.
$$

Since $r : [0, \infty) \to [0, \infty)$ is an increasing function satisfying $r(0) = 0$, $r(t) > 0 \ \forall \ t > 0$. Hence, we have

$$
\lim_{n \to \infty} d(a_n, F_{ix}(\varphi)) = 0.
$$

Now, all the conditions of Theorem 3 are satisfied; therefore, $\{a_n\}$ converges strongly to a fixed point of $\varphi$. $\square$

In Banach spaces with an Opial condition, we had a weak convergence of our iterative algorithm. However, if the mapping satisfied Condition (I), then we obtained a strong convergence result.

## 4. Numerical Example

**Example 2.** *Let $G = [0, \infty)$ be endowed with usual norm $|.|$. Let $\varphi : [0, \infty) \to [0, \infty)$ be defined by*

$$\varphi(a) = \begin{cases} \frac{a+1}{2} & \text{if } a \geq 3 \\ 0 & \text{if } a \in [0, 3). \end{cases}$$

*$\varphi$ does not satisfy Condition (C). Moreover, $\varphi$ is generalized $\alpha$-nonexpansive mapping. Let $a = \frac{5}{2}$ and $b = \frac{7}{4}$ since we have $\varphi(a) = \frac{7}{2}$, so,*

$$\frac{1}{2}|a - \varphi(a)| = \frac{1}{2}|\frac{5}{2} - \frac{7}{2}| = \frac{1}{2}|\frac{3}{4}| = \frac{3}{8}.$$

*In addition, $|a - b| = |\frac{5}{2} - \frac{7}{4}| = 1$.*
*So,*

$$\frac{1}{2}|a - \varphi(a)| < |a - b|,$$

*but $|\varphi(a) - \varphi(b)| > |a - b|$. Hence, $\varphi$ does not satisfy Condition (C).*
*Now, taking $\alpha = \frac{1}{3}$, consider the following cases.*
*Case 1: If $a > 3$ and $b \in [0, 3]$, then*

$$|\varphi(a) - \varphi(b)| = |\frac{a+1}{2} - 0| = \frac{1}{2}|a + 1|.$$

*In addition,*

$$\alpha|b - \varphi(a)| + \alpha|a - \varphi(b)| + (1 - 2\alpha)|a - b| = \frac{1}{3}|b - \frac{a+1}{2}| + \frac{1}{3}|a| + \frac{1}{3}|a - b|$$

$$\geq \frac{1}{2}|a + 1| = |\varphi(a) - \varphi(b)|.$$

*Case 2: For $a > 3$ and $b > 3$, we have*

$$|\varphi(a) - \varphi(b)| = |\frac{a+1}{2} - \frac{a+1}{2}| = \frac{1}{2}|a - b|$$

*and*

$$\alpha|b - \varphi(a)| + \alpha|a - \varphi(b)| + (1 - 2\alpha)|a - b| = \frac{1}{3}|b - \frac{a+1}{2}| + \frac{1}{3}|a - \frac{b+1}{2}| + \frac{1}{3}|a - b|$$

$$\geq \frac{1}{2}|a - b| = |\varphi(a) - \varphi(b)|.$$

*Case 3: Let $a \in [0, 3]$ and $b > 3$. Then,*

$$\frac{1}{3}|b - \varphi(a)| + \frac{1}{3}|a - \varphi(b)| + \frac{1}{3}|a - b| \geq |\varphi(a) - \varphi(b)|.$$

*Hence, $\varphi$ is generalized $\alpha-$nonexpansive mapping.*

We now present an experiment to compare the convergence behavior of iteration (5). Take initial values $a_1 = 8 \in G$ and $\eta_n = \frac{n}{n^2+4n+2}$, $\rho_n = \frac{n+1}{n^2+n+1}$ and $\sigma_n = \frac{2n+1}{n^2+n+7}$. Iterative Algorithm (5) converged faster than the other schemes for generalized $\alpha$-nonexpansive mapping (Figure 1).

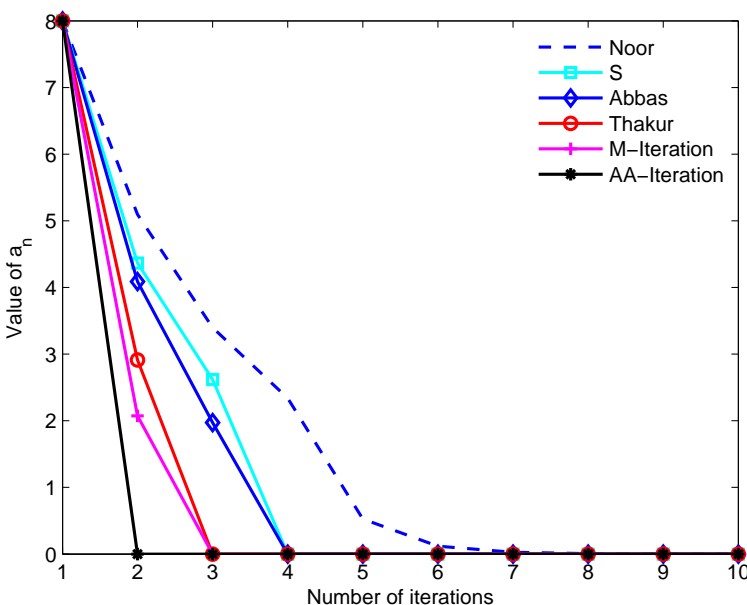

**Figure 1.** Convergence behavior of iterative algorithms.

## 5. Application

In 2022, El-sayed and Omar [32] established the existence and uniqueness of the weak solution of a delay composite functional differential equation of the Volterra–Stieljes type. Many authors solved the delay composite functional differential equation of the Volterra–Stieljes type. For more details, we refer to [33,34]. In this section, we estimate the weak solution of a delay composite functional differential equation.

Let $B$ be a reflexive Banach space with norm $\|.\|_B$, $B^*$ denotes the dual of $B$ and $C[J, B], J = [0, M]$ denotes the class of continuous functions equipped with the following norm:

$$\|a\|_C = sup_{t \in J}\|a(t)\|_B, \ a \in C[J, B].$$

Consider the following delay composite functional differential equation of the Volterra–Stieltjes type:

$$\frac{d}{dt}a(t) = f_1\left(t, \int_0^{h(t)} f_2(t, s, a(s))d_s g(t, s)\right), \ t \in J \tag{39}$$

with initial condition

$$a(0) = a_0, \tag{40}$$

Assume that

(i). $h : J \to J$ is continuous increasing with $h(t) \le t$ .

(ii). $f_1 : J \times B \to B$ is weakly continuous and satisfies the weak Lipschitz condition with Lipschitz constant $L_1$, such that

$$\left|T(f_1(t, a)) - f_1(t, b)\right| \le L_1\left|T(a - b)\right|, \ L_1 > 0, \forall (t, a), (t, b) \in J \times B, T \in B^*.$$

(iii). $f_2 : J \times J \times B \to B$ is weakly continuous and weakly satisfies the Lipschitz condition with Lipschitz constant $L_2$ such that

$$\left|T(f_2(t, s, a)) - f_2(t, s, b)\right| \le L_2\left|T(a - b)\right|,$$

(iv). Function $g : J \times \mathbb{R} \to \mathbb{R}$ is continuous with

$$w = \max\left\{sup\left|g(t, h(t))\right| + sup\left|g(t, 0)\right|\right\} \text{ on } J.$$

(v). $L_1 L_2 wt \ < 1$

Finding the solution of (39) and (40) is equivalent to finding the solution of the following integral equation [32]:

$$a(t) = a_0 + \int_0^t f_1\left(s, \int_0^{h(t)} f_2(s, \theta, a(\theta))\right) d_\theta g(s, \theta) ds.$$

In the following theorem, we obtain an approximation of the solution of (39) and (40) using $AA$-iterative Algorithm (5).

**Theorem 6.** *Suppose that Assumptions (i)–(v) hold. Then, Problems (39) and (40) have a unique solution $a^* \in C[J, B]$, and the sequence $\{a_n\}$ defined in (5) converges to $a^*$.*

**Proof.** Let $\{a_n\}$ be a sequence defined in (5). Define an operator $\varphi$ on $C[J, B]$ by

$$\varphi(a(t)) = a_0 + \int_0^t f_1\left(s, \int_0^{h(t)} f_2(s, \theta, a(\theta))\right) d_\theta g(s, \theta) ds.$$

Note that,

$$
\begin{aligned}
\|d_n - a^*\|_C &= \|(1 - \sigma_n)a_n + \sigma_n \varphi(a_n) - a^*\|_C \\
&\leq (1 - \sigma_n\|a_n - a^*\| + \|\varphi(a_n) - a^*\|_C
\end{aligned}
\tag{41}
$$

and

$$
\begin{aligned}
\|\varphi(a_n) - a^*\|_C &= \|\varphi(a_n) - \varphi(a^*)\| \\
&\leq \left\| a_0 + \int_0^t f_1\left(s, \int_0^{h(s)} f_2(s, \theta, a_n(\theta))\right) d_\theta g(s, \theta) ds - \right. \\
&\qquad \left. a_0 - \int_0^t f_1\left(s, \int_0^{h(s)} f_2(s, \theta, a^*(\theta))\right) d_\theta g(s, \theta) ds \right\|_C \\
&= \left| \varphi\left(\int_0^t f_1\left(s, \int_0^{h(s)} f_2(s, \theta, a_n(\theta))\right) d_\theta g(s, \theta) ds - \right.\right. \\
&\qquad \left.\left. \int_0^t f_1\left(s, \int_0^{h(s)} f_2(s, \theta, a^*(\theta))\right) d_\theta g(s, \theta)\right) ds \right| \\
&\leq \int_0^t L_1 \left| \varphi\left(\int_0^{h(s)} f_2(s, \theta, a_n(\theta)) ) d_\theta g(s, \theta) - \right.\right. \\
&\qquad \left.\left. \int_0^{h(s)} f_2(s, \theta, a^*(\theta)) d_\theta g(s, \theta)\right) \right| ds \\
&\leq L_1 \int_0^t \int_0^{h(s)} \left| \varphi\left(f_2(s, \theta, a_n(\theta))\right) - \right. \\
&\qquad \left. f_2(s, \theta, a^*(\theta))\right) d_\theta g(s, \theta) \right| ds \\
&\leq L_1 \int_0^t \int_0^{h(s)} L_2 \left| \varphi\left(a_n(\theta) - a^*(\theta) d_\theta g(s, \theta)\right) \right| ds \\
&= L_1 L_2 \|a_n - a^*\|_C \int_0^t \int_0^{h(s)} d_\theta g(s, \theta) ds \\
&= L_1 L_2 \|a_n - a^*\|_C \int_0^t (g(s, h(s)) - g(s, 0)) ds \\
&\leq L_1 L_2 \|a_n - a^*\|_C \int_0^t ds \\
&= L_1 L_2 w t \|a_n - a^*\|_C \\
&\leq \|a_n - a^*\|_C
\end{aligned}
\tag{42}
$$

So,

$$\begin{aligned}
\|d_n - a^*\|_C &\leq (1 - \sigma_n)\|a_n - a^*\|_C + \sigma_n\|a_n - a^*\|_C \\
&\leq \|a_n - a^*\|_C
\end{aligned} \tag{43}$$

Now, let $e_n = (1 - \rho_n)d_n + \rho_n\varphi(d_n)$; following the arguments similar to those given above, we obtain

$$\|e_n - a^*\|_C \leq \|d_n - a^*\|_C \leq \|a_n - a^*\|_C.$$

So,

$$\|c_n - a^*\|_C \leq \|\varphi(c_n) - \varphi(a^*)\|_C.$$

From (42), we obtain

$$\|c_n - a^*\|_C \leq \|e_n - a^*\|_C \leq \|a_n - a^*\|_C.$$

Similarly,

$$\|b_n - a^*\|_C \leq \|a_n - a^*\|_C.$$

If we set $\|a_n - a^*\|_C = v_n$, then we obtain

$$v_{n+1} \leq v_n, \quad \forall \quad n \in \mathbb{N}$$

which implies that

$$\lim_{n \to \infty} v_n = 0.$$

Hence, $a_n \to a^*$. □

## 6. Conclusions

In this paper, we approximated the fixed points of generalized $\alpha$-nonexpansive mappings using an $AA$-iterative algorithm. We established some weak and strong convergence results for generalized $\alpha$-nonexpansive mappings in uniformly convex Banach spaces. A numerical example was given to show that $AA$-iterative algorithm converged faster than some existing algorithms for generalized $\alpha$-nonexpansive mappings. We approximated the weak solution of delay composite functional differential equation of the Volterra–Stieltjes type by $AA$-iterative scheme. In future work, we shall extend these results for some general class of mappings in some important abstract spaces, and try to extend the iterative scheme to approximate the solution of certain nonlinear problems, such as fixed-point and optimization problems in fewer steps.

**Author Contributions:** I.B., M.A. and M.W.A. contributed to the study conception, design, and computations. M.W.A. wrote the first draft of the manuscript, and all authors commented on it. All authors have read and approved the final manuscript.

**Funding:** This research received no external funding.

**Institutional Review Board Statement:** Not applicable.

**Informed Consent Statement:** Not applicable.

**Data Availability Statement:** Not applicable.

**Acknowledgments:** The authors are grateful to the reviewers for their useful comments, which helped to improve the presentation of this paper.

**Conflicts of Interest:** The authors declare that they have no conflict of interest.

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
