# Peer review of "Convergence of AA-Iterative Algorithm for Generalized α-Nonexpansive Mappings with an Application"

_mathematics, doi:10.3390/math10224375_

Round 1

Reviewer 2 Report

The authors present new interesting and significant results showing how to approximate the fixed points of generalized apha-nonexpansive mappings using certain iterative algorithm, called by them AA-iterative algorithm. They prove some some weak and strong convergence results in uniformly convex Banach spaces. They also provide a numerical example showing that the AA-iterative algorithm converges faster than some other algorithms and present an approximation of the weak solutions of a delay composite functional differential equation of Volterra-Stieltjes type. Some of the results generalize corresponding outcomes of Thakur and Ali. Moreover, the authors suggest some issues that can be studied in the future.

In general the paper is clearly written, but some corrections are necessary. For example:

1. In the second line of Introduction I suggest to delete "a Banach space", because this is written already in the first line.

2. The last line on page 1 I suggest to rewrite, e.g., as follows: "hence condition (C) is weaker than the one depicting nonexpansive mappings. However,"

and the first line on page 2 should start. e.g., with: "condition (C) is stronger than the one defining quasi-nonexpansive mappings."

Similar mistakes occur in the whole paper and should be corrected in a similar way.

Reviewer 3 Report

See pdf file

Reviewer 4 Report

The article I have been asked to evaluate is part of the research theme, inaugurated by Suzuki's famous articulation, which in the submitted manuscript is the entry [31], on the existence of fixed points in the case of non-expansive operators. In contrast to many contributions in this area, which in my opinion have a sterile abstract approach, in this paper the authors propose explicit iterative methods for finding solutions, also presenting useful concrete examples. The exposition is clear and the use of English is appropriate to the level of an important journal such as Mathematics. Note that there is probably a misprint in formula (5.1): there is an unclosed parenthesis
